# Randomised controlled feasibility trial of standard wound management versus negative-pressure wound therapy in the treatment of adult patients having surgical incisions for hip fractures

James P M Masters,[1] Juul Achten,[1] Jonathan Cook,[2] Melina Dritsaki,[3] Lucy Sansom,[1] Matthew L Costa[1]

[1]Oxford Trauma, Nuffield Department of Orthopaedics, Rheumatology and Musculoskeletal Sciences (NDORMS), University of Oxford, Oxford, UK
[2]Centre for Statistics in Medicine, Nuffield Department of Orthopaedics, Rheumatology and Musculoskeletal Sciences (NDORMS), University of Oxford, Oxford, UK
[3]Oxford Clinical Trial Unit, Nuffield Department of Orthopaedics, Rheumatology and Musculoskeletal Sciences (NDORMS), University of Oxford, Oxford, UK

**Correspondence to**
Dr James P M Masters;
james.masters@ndorms.ox.ac.uk

## ABSTRACT

**Introduction** Deep wound infection is a catastrophic complication after hip fracture surgery. However, current understanding of infection rates in this population is limited. Many technologies such as incisional negative-pressure wound therapy (NPWT) show promise in reducing the rate of infection. This trial is a feasibility study looking to establish a value estimated with a greater precision of the rate of deep infection after hip fracture treatment in patients treated with NPWT versus standard dressing following hip fracture surgery.

**Methods and analysis** A randomised controlled trial of 464 patients will be run across multiple centres. It is embedded in the World Hip Trauma Evaluation cohort study. Any patient over the age of 65 years having surgery for hip fracture is eligible unless they are being treated with percutaneous screw fixation. A web-based randomisation sequence will stratify patients by centre. Patients will be allocated to either NPWT or standard care on a 1:1 basis. The primary outcome measure is the Centre for Disease Control definition of deep infection at 30 days. Follow-up at 4 months will also assess deep infection and the core outcome dataset for hip fractures. This includes health-related quality of life (EQ-5D-5L), mobility, mortality and late complications such as further surgery. The primary analysis will be intention to treat.

**Ethics and dissemination** Oxford C Research Ethics Committee granted ethical approval on 28/04/2017, 17/SC/0207. The results of this study will be reported in a peer-reviewed publication and inform the design of a future full-scale trial.

**Trial registration number** ISRCTN55305726.

## Strengths and limitations of this study

► This prospective trial will estimate the true rate of infection according to nationally recognised criteria.
► Robust data collection systems will be used to determine the rate of deep infection.
► This study will include the full range of hip fracture patients including those without capacity, across multiple centres ensuring good external validity to our findings.
► The potential limitation will be to accurately capture those cases of infection where patients move through different institutions and care pathways.

cost of this clinical problem is estimated at 1.75 million disability-adjusted life years lost and represents 1.4% of the total healthcare burden in established market economies.[4]

The overwhelming majority of hip fractures are treated surgically. The principal objective of surgical treatment is to enable patients to return to weight-bearing mobilisation as soon as possible.

As with all surgical treatments, one of the most important potential complications is surgical site infection (SSI). Hip fracture patients are particularly vulnerable to SSI. The reasons for this vulnerability are multifactorial. For instance, 60% of hip fracture patients have at least one major medical comorbidity such as steroid or other immunosuppressive medication use, malignancy and diabetes.[5] These comorbidities sit alongside age-related deterioration of immune function.

SSI can be either superficial or deep. Superficial infections only involve the skin and subcutaneous tissue, whereas 'deep' infections involve the deeper layers and the metalwork implanted in the hip. Deep infection

## INTRODUCTION

Hip fracture is one of the biggest challenges facing patients and healthcare systems at present. Worldwide, there are 1.3 million hip fractures with more than 65 000 hip fractures in England, Wales and Northern Ireland every year.[1] These figures are projected to rise to >100 000 by 2020 in the UK[2] and more than 6 million by 2050 worldwide.[3] The global

has profound consequences for hip fracture patients including a significantly increased hospital stay and they are 4.5 times less likely to survive to discharge.[6] Of those that do survive, patients with deep infection have been found to be three times less likely to return to their original residence.[6] These findings have been found in other large cohort studies.[7]

Due to the severe consequences of deep infections in this patient group, a number of interventions are undertaken to try and reduce the risk of infection such as perioperative antibiotics, vigorous skin preparation and wound dressing. Wound dressing is one of the most important interventions to address the risk of postoperative infection. The role of a postoperative wound dressing is to minimise bacterial ingress into the wound, manage exudate from the wound site and protect the soft-tissues while they heal.

Traditionally, the surgical incision is covered with an adhesive dressing or gauze to protect the wound from contamination from the outside environment. These 'standard dressings' have been used throughout the National Health Service (NHS) for many years. Negative-pressure wound therapy (NPWT) is an alternative form of dressing which may be applied to closed surgical incisions. In this treatment, an 'open-cell', solid foam overlies the incision and is covered with a semipermeable membrane which is only permeable to gas. A sealed tube is used to connect the foam to a pump, which creates a partial vacuum over the wound. This negative-pressure therapy provides a sealed environment, preventing bacterial ingress and removes blood and serous fluid exuding the wound. The application of negative pressure to the foam leads to the application of positive pressure to the wound bed and has been shown to reduce the incidence of wound haematoma.[8] Recent laboratory studies suggest that NPWT shifts the cytokine profile to being less inflammatory, promotes the production of proangiogenic growth factors and enzymes responsible for matrix remodelling, leading to improved wound healing.[9–13]

However, a recent Cochrane review for surgical wounds concluded, 'it is still not clear whether NPWT promotes faster healing and reduces complications associated with clean surgery'. 'Given the cost and widespread use of NPWT, there is an urgent need for suitably powered, high-quality trials to evaluate the effects of the newer NPWT products that are designed for use on clean, closed surgical incisions. Such trials should focus initially on wounds that may be difficult to heal'.[14]

The current feasibility trial will assess the feasibility of performing a randomised controlled trial (RCT) on wound dressings in this challenging population, and it will aim to estimate the current infection rates for hip fracture patients. Capture of SSI rates falls under the remit of Public Health England's Surgical Site Infection Surveillance Programme.[15] Their report from 2015 reports a SSI incidence of 1.3% (95% CI 1.3% to 1.5%).[15] However, infection rates reported in the literature for this patient group are often higher, with some reports stating rates as high as 7%.[16 17] There is growing evidence that the surveillance system from Public Health England is providing a significant underestimate of the true incidence.[18–20] This is borne out by local audit of hip fracture patients at two large teaching hospitals in the England, that have found infection rates between 5% and 8%, respectively (unpublished data). A reliable estimation of infection will be required to inform the sample size for a large RCT.

## AIMS AND OBJECTIVES

We aim to conduct a feasibility trial of NPWT versus standard dressing following hip fracture surgery. It is intended to obtain a reliable estimate of the rate of SSIs in the target population. In addition, feasibility will be considered in terms of other aspects of the trial such as recruiting and randomising large numbers from individual sites within a relatively short time frame.

The primary objective is:

To quantify differences in the rate of 'deep infection'—as defined by the Centre for Disease Control (CDC)—after hip fracture surgery in standard dressing and NPWT. Patients will be assessed at 30 days and 4 months. We will examine how many deep infections occur, and when the diagnosis is made.

Both 'how many deep infections' and 'when they are diagnosed' will be crucial to inform the design of a full-scale trial. Each time point addresses the requirements of the CDC, and the routine follow-up of patients in the current UK care pathway.

The secondary objectives are:
1. To determine the number and nature of further surgical interventions related to the injury, in the first 4 months after the index procedure.
2. To investigate which outcomes will be required to assess cost-effectiveness, of NPWT versus standard dressing for wounds associated with hip fracture surgery in a definitive trial.
3. To determine recruitment rate and willingness to participate in the trial.
4. To capture the core outcome set for hip fracture studies.

## METHODS AND ANALYSIS
### Study design
This is a multicentre two arm non-blinded randomised feasibility study embedded within a prospective cohort.

### Setting and participants
This feasibility study is embedded within the World Hip Trauma Evaluation cohort.[21] It will be run in five different centres across England including Major Trauma Centres and Trauma Units. All patients being surgically treated for hip fracture will be considered for enrolment.

### Eligibility
Patients will be eligible for this study if
▶ They are aged 65 years or older.

► They have a hip fracture requiring surgical treatment.

Exclusion criteria: Patients having percutaneous screw fixation of an undisplaced intracapsular fracture of the hip. This small subgroup of patients (less than 5% on the National Hip Fracture Database (NHFD)) have very small surgical incisions—typically 1–2 cm. Where incisions are so small, it is unlikely that they will benefit from an advanced dressing like NPWT.

## Recruitment of participants, screening and eligibility assessment

### Consent

Patients with a hip fracture are a clinical priority for urgent operative care. They will undergo surgery on the next available trauma operating list. All patients with a fracture of the hip are in pain and have received opiate analgesia. It is therefore understandable that the majority of patients find the initial period of their treatment in hospital confusing and disorientating. Similarly, patients' next of kin, carers and friends are often anxious at this time and may have difficulty in weighing the large amounts of information that they are given about the injury and plan for treatment. In this emergency situation, the focus is on obtaining consent for surgery (where possible) and informing the patient and any next of kin about immediate clinical care. This is supported by national guidance that recommends surgery should take place within 36 hours.[22] It is often not possible for the patient or relative/carer (consultee) to review trial documentation, weigh the information and communicate an informed decision about whether they would wish to participate. The consent procedure for this trial will reflect that of the surgery, with the clinical team assessing capacity before taking consent for the surgical procedure and this capacity assessment then being used to decide on the proper approach to consenting to the research. An appropriate method, in line with the mental capacity act and approved by a National Research Ethics Committee, will then be used to gain either prospective or retrospective consent from the patient or appropriate consultee by a Good Clinical Practice (GCP)-trained, appropriately delegated member of the research team.

## Randomisation

The treating surgeon will confirm eligibility at the end of the operative procedure, but before the wound dressing is applied. Eligible patients will be enrolled into the study via a secure, remote online randomisation system. To ensure concealment of the allocation sequence, patients will only be randomised after they have been registered on the online system. The allocation sequence will be generated by the trial statistician at the Oxford Clinical Trials Unit. Randomisation will be on a 1:1 basis, stratified by trial centre. All modern operating theatres include a computer with web access, so a secure, 24 hours, web-based randomisation system will be used to generate the treatment allocation intraoperatively.

## Postrandomisation withdrawals/exclusions

Participants may decline to continue to take part in the trial at any time without prejudice. A decision to decline consent or withdraw will not affect the standard of care the patient receives.

## Blinding

As the wound dressings are clearly visible, the patients cannot be blind to their treatment. In addition, the treating surgeons will also not be blind to the treatment, but will take no part in the postoperative research assessment of the patients. The clinical outcome data will be collected by independent research assistants who will not be blind to the treatment allocation. Questionnaire data will be entered onto the trial central database by a data clerk in the trial central office.

## Interventions

Patients with a hip fracture usually have surgery on the next available trauma operating list. Some patients may be delayed by medical factors such as anticoagulation, however, surgery will proceed as soon as is determined safe by the treating team. All patients will receive a general or regional anaesthetic. Routine prophylactic antibiotics will be given to all patients immediately before surgery and all patients will be assessed for venous thromboembolism prophylaxis according to local policy. At the end of the operation, a dressing is applied to the surgical wound. This trial will involve two types of wound dressing: standard dressing and NPWT.

### Standard dressing

The standard dressing for a surgical wound comprises a non-adhesive layer applied directly to the wound that is covered by a sealed dressing. The standard dressing does not use 'negative pressure'. The exact details of the dressing materials and duration will be left to the discretion of the treating surgeon as per their routine practice, but the details of each dressing applied in the trial will be recorded.

### Negative-pressure wound therapy

The NPWT dressing uses an 'open-cell', solid foam layer which is laid onto the wound as an intrinsic part of a sealed dressing. A sealed tube connects the dressing to a built in mini-pump that creates a partial vacuum over the wound. This can only be kept on for 7 days according to the manufacturer that will be reflected in this study.

Any further wound dressing after the initial dressing will be recorded and will follow the allocated treatment unless otherwise clinically indicated.

## Outcome measures

### Primary outcome

The primary outcome measure for this study is deep infection; We will use the CDC and Prevention definition of a 'deep SSI', that is a wound infection involving the tissues deep to the skin that occurs within 30 or 90 days of injury.[23]

The treating clinical team will make the diagnosis of 'deep infection', as per routine clinical practice. The treating clinicians will not be part of the research team. Since the prompt diagnosis and treatment of infection is fundamental to the patient's routine clinical care, the treating surgeon/clinician will always document such a change in management in the patient's medical record.

In addition to the rate of infection, time of diagnosis will be assessed; The CDC definition applies up to the 90 days after hip fracture surgery.[23] However, the national care pathway for hip fractures involves remote follow-up at 4 months. Therefore, to ensure that assessment meets the CDC criteria and is consistent with existing care practice, each patient will be assessed at 30 days and 4 months.

At 30 days, the patient will be assessed either as an inpatient or over the telephone according to the relevant clinical parameters.

At 4 months patients will be asked, via telephone interview, about any further symptoms of infection which may have emerged since the 30-day time point. Of the infections that are identified, we will establish when the diagnosis was made in the course of recovery.

### Secondary outcome measures in this trial
#### Mortality
Qualitative work with patients who sustain hip fractures identified mortality as an important metric.[24]

#### EuroQol EQ-5D-5L
The EuroQol EQ-5D is a validated measure of health-related quality of life, consisting of a five-dimension health status classification system and a separate Visual Analogue Scale.[25] Responses to the health status classification system will be converted into multiattribute utility (MAU) scores using tariffs currently under development for England.[26] These MAU scores will be combined with survival data to generate Quality Adjusted Life Year (QALY) profiles for the purposes of the economic evaluation. The EQ-5D has been validated to be completed by a patient's proxy in case of continues impaired capacity.

#### Complications
All complications and surgical interventions related to the index wound will be recorded.

#### Resource use
Will be monitored to help inform the economic analysis plan in the subsequent definitive trial. Cost data will be obtained from national databases or will be estimated in consultation with the hospital finance department. The cost consequences following discharge, including NHS costs and patients' out-of-pocket expenses will be recorded via a short questionnaire, which will be administered at 4 months postinjury. This will be either by patient or consultee.[27]

#### Mobility
The ability to walk indoors and outdoors is rated very highly by patients. It has been included in a recommended 'core outcome set' for trials assessing interventions in hip fractures.

#### Residential status
Also captured on case report form (CRF). The residential status is also part of the core outcome set for hip fractures and NHFD dataset.[24] It will be captured at baseline and at 4 months.

#### Recruitment rate
The rate of recruitment of potential participants, both in terms of eligibility and those who consent (or on whose behalf consent is provided) will be recorded.

#### Retention rate
The rate of retention through the study visits of participants will be recorded.

### Adverse event management
Adverse events (AEs) are defined as any untoward medical occurrence in a clinical trial subject and which do not necessarily have a causal relationship with the treatment. All AEs will be listed on the appropriate CRF for routine return to the 'WHISH' central office.

Some AEs are expected as part of the surgical interventions, and do not need to be reported immediately, provided they are recorded in the 'Complications' section of the CRFs and/or patient questionnaires. These events are: complications of anaesthesia or surgery (bleeding or damage to adjacent structures such as nerves, tendons and blood vessels, delayed unions/non-unions, further surgery to remove/replace metalwork, dislocation, wound dehiscence and thromboembolic events). All participants experiencing Significant Adverse Events (SAEs) will be followed up as per protocol until the end of the trial.

### Follow-up
Each patient will be assessed in the current UK national audit framework for hip fracture patients. This means they will contact by telephone 4 months after surgery. As per the feasibility aims of this study, we will also assess each patient at 30 days. This may be in person as an inpatient or over the telephone is the patient has been discharged.

The 4-month time point will collect the data required for the NHFD, health-related quality of life and the other components of the hip fracture core outcome set.[24] We will also investigate deep infection at the 4-month time point to ensure that each patient has been assessed by 90 days which is required by the CDC.

Details of any late complications will be sent securely to the trial coordinating centre. Complications will be captured on the CRF by the site research associate and recorded on the CRFs at 30 days/discharge and at 4 months via telephone interview.

Where a personal consultee has agreed to the patient's involvement, we will ask them for the follow-up data if the participant has permanently lost capacity to consent. Those patients or consultees who do not wish to provide follow-up data but agree to the study team reviewing

their medical records, clinical reporting forms will be completed by the site and returned in the same manner. This is referred to as 'routine data only' option.

## Sample size

The purpose of the proposed feasibility study is to determine the rate of deep SSI which in turn will inform the sample size for the definitive trial. We will recruit a sample of 464 participants; divided equally in each of the two study arms. This is based on the number of hip fractures treated at each of the recruiting centres and assuming a conservative recruitment of 50% of eligible patients. Four hundred and sixty-four is based on Wilson's CI method at two-sided 95% significance level with 20% loss to follow-up). This will allow us to estimate the overall 30-day deep infection rate within 3%–5% depending on the actual rate.

## Rehabilitation

Early full weight-bearing mobilisation is routine in all cases of hip fracture. Further details of the rehabilitation will be left to the discretion of the treating team as per usual clinical practice.

## Data management

The CRFs will be designed by the trial manager in conjunction with the trial management team. All electronic patient-identifiable information will be held on a secure, password-protected database at the Kadoorie Centre, accessible only to the research team. Paper forms with patient-identifiable information will be held in secure, locked filing cabinets within a restricted area. Patients will be identified by a code number only. Direct access to source data/documents will be required for trial-related monitoring and/or audit by the Sponsor, NHS Trust or regulatory authorities as required. All paper and electronic data will be retained for at least 5 years after completion of the trial.

## Statistical analysis

Standard statistical summaries (eg, means, SDs, medians, IQR) will be reported for all discrete and continuous outcome measures. Baseline data (eg, age and gender) will be summarised to check comparability between treatment arms. This will only be shared with the Data Management Committee (DMC) in the first instance at the end of the feasibility study. No formal hypothesis testing will take place; as per the aim of this feasibility study, the analysis will report the deep infection rates overall and in the two treatment groups on an intention-to-treat basis at 30 days and 4 months postrecruitment.

Should this study demonstrate that a full-scale trial is feasible, the data from this feasibility study will be included in the final analysis of the definitive trial. This is contingent on the future definitive trial being sufficiently similar to this feasibility study. Baseline and follow-up outcome data will therefore not be released to the trial team unless the lack of feasibility is determined based on observed infection rate at the end of the feasibility study,

or the definitive study cannot be conducted for other reasons (eg, lack of funding). Reporting of the study outcomes will be done accordingly to enable to use of the outcome data in a future definitive trial if such a study goes ahead.

Recruitment and retention rates will be reported through a consort diagram and so will inform the planning of a definitive trial with respects to sample size and the time points of measurements.

It seems likely that some data may not be available due to voluntary withdrawal of patients, lack of completion of individual data items or general loss to follow-up. In this hip fracture cohort, there is significant mortality at 30 days—6.7%.[1] Where possible the reasons for missing data will be ascertained and reported. Although missing data is not expected to be a problem for this study, the nature and pattern of the missing data is important to understand for the full study, so will be carefully considered.

Reasons for ineligibility, non-compliance, withdrawal or other protocol violations will be stated and any patterns summarised.

## Economic evaluation

An economic evaluation will not be the primary focus of the trial as it is a feasibility study. However, we will seek to collect relevant health economic data to explore how well this data can be reported in this group and whether any adaptations need to be undertaken to facilitate this. Given the sizeable financial difference between the two proposed interventions, a clear understanding of the health economics will be imperative in any subsequent larger trial.

## Trial oversight

The day-to-day management of the trial will be the responsibility of the Clinical Trial Manager, based at Nuffield Department of Orthopaedics, Rheumatology and Musculoskeletal Sciences and supported by the Oxford Clinical Trials Research Unit (OCTRU) staff. This will be overseen by the Trial Management Group, who will meet monthly to assess progress. It will also be the responsibility of the Clinical Trial Manager to undertake training of the research associates at each of the trial centres. The Trial Statistician and Health Economist will be closely involved in setting up data capture systems, design of databases and clinical reporting forms.

A Trial Steering Committee (TSC) and a Data and Safety Monitoring Committee (DSMC) will be set up. The study DSMC will adopt a (DA ta MO nitoring C ommittees: L essons, E thics, S tatistics) DAMOCLES charter which outlines its responsibilities. They will not perform any formal interim analyses of effectiveness. They will see copies of data accrued to date, or summaries of that data by treatment group. They will also assess the screening algorithm against the eligibility criteria. Emerging evidence from other related trials or research will be considered. Significant AEs will be reviewed. The DSMC

may tell the chair of the TSC at any time if, in their view, the trial should stop for ethical reasons. This includes concerns about participant safety. The DSMC will meet at least once per year during the recruitment phase of the study.

## Quality control

The study may be monitored, or audited in accordance with the current approved protocol, relevant regulations and standard operating procedures by the Host organisation, Sponsor or appropriate Regulatory Authorities. A Monitoring Plan will be developed according to OCTRU standard operating procedures which involves a risk assessment. The monitoring activities are based on the outcome of the risk assessment and may involve central monitoring and site monitoring.

## Ethics and dissemination

Results of this study will be disseminated through peer-reviewed publication and presentation at conferences.

**Contributors** JPM, MLC and JA wrote the background section and developed the research question. JPM, MLC, JA and LS were responsible for the research methodology and management sections of the protocol. JC wrote the sample size and statistical analysis sections of the protocol. MD wrote the health economic evaluation section of the protocol. All authors reviewed and agreed the final manuscript.

**Funding** The study is funded by the Royal College of Surgeons of England and The Dunhill Medical Trust [DMT/RCS605] . It is supported by the National Institute for Health Research (NIHR) Oxford Biomedical Research Centre. Excess treatment costs for the negative pressure wound therapy arm of the study are met by Smith and Nephew.

**Competing interests** None declared.

**Patient consent** Not required.

**Ethics approval** Oxford C Research Ethics Committee granted ethical approval on 28/04/2017, 17/SC/0207.

**Provenance and peer review** Not commissioned; externally peer reviewed.

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
