## [Reviewer comments · BMJ Open]

ARTICLE DETAILS

TITLE (PROVISIONAL)	WHITE 7 - WHISH – Wound Healing in Surgery for Hip fractures A Randomised Controlled Feasibility Trial of standard wound management versus negative pressure wound therapy in the treatment of adult patients having surgical incisions for hip fractures
AUTHORS	Masters, James; Achten, Juul; Cook, Jonathan; Dritsaki, Melina; Sansom, Lucy; Costa, Matthew

VERSION 1 – REVIEW

REVIEWER	Michael Whitehouse Musculoskeletal Research Unit, Translational Health Sciences, Bristol Medical School, University of Bristol, UK
REVIEW RETURNED	10-Dec-2017

GENERAL COMMENTS	Thank you for the opportunity to review this trial protocol in an important area of clinical research. The study would provide valuable information to optimise the care of this complex patient group. I would prefer to see a standard consent form included as well as the current "Prospective Confirmation of Informed Agreement" form as patients capable of providing informed consent should be given the opportunity to do so and the wording of the form does not really fit for this group. There are two full stops at the end of the first bullet point on strengths and limitations of the study (page 10 of the submission pdf). The spacing between paragraphs (no line space between paragraph 1 and 2) and line spacing (e.g. single line spacing in last paragraph) are not consistent in the introduction. The full stop is missing from the end of the 3rd paragraph of the introduction. The spacing between the text and citations is not consistent, there is a space in some cases and not in others. Later in the document, some citations appear after the punctuation rather than before as elsewhere in the text. The full stop is missing from the end of the first paragraph of the aims and objectives section. The full stop is missing from the end of the numbered list of secondary objectives. There is also not one at the end of the first numbered point but there is at the end of the second and third, please be consistent in the use of punctuation. In the study design, define that the cohort is prospective. Cite the latest NHFD report when data from it is cited. The style of text is different in the recruitment section to that used previously. As the document of review is a pdf I am not sure if this is just the text font or size as well but please format consistently
---

	throughout. Later on, non-justified text is used whereas it is justified earlier on. When discussing the urgency of operative care, please mention the current NICE target of surgery within 36 hours and cite the guidance. Blinding - there is a space missing in the last sentence ("...central database by adata clerk in the..."). It should be acknowledged that the CDC definition of deep infection in the presence of an implant takes place at 1 year following surgery and therefore in this feasibility study, an earlier time point has been chosen and this decision justified. There is inconsistent use of semicolons after the bold text in the list of secondary outcome measures on pages 16-17. Adverse event management: The classification of "further surgery to remove/replace metalwork" and "wound dehiscence" as complications that are expected as part of the surgical intervention in the context of a trial with infection as the primary outcome measure is interesting as both of these complications will be strongly correlated with the primary outcome. Whilst I agree with the reporting time line suggested by the authors, careful consideration needs to be given to whether these outcomes are included as primary outcomes, adverse events or complications. Consideration also needs to be given to the classification of the outcomes of interest given this is a feasibility study design. In the context of a feasibility study, it is usual to consider outcomes such as the ability to recruit to the trial, recruitment rates, completion of outcome measures etc. as the outcomes of the study. The primary outcome described (deep infection) is actually the primary outcome of the definitive study proposed. The "Rehabilitation" subtitle on page 18 is indented whereas none of the other subtitles are. Statistical analysis - the inclusion of data collected in this feasibility study in the analysis of any subsequent definitive trial suggests that the authors intend to use exactly the same outcome measures. This would be more consistent with this being a pilot study design. This also further emphasises the critical need to justify the decision not to use the standard CDC definition of deep infection in the presence of an implant at 1 year post intervention. The formatting of the references is not consistent, particularly the amount of information provided and the formatting of dates.
--	---

REVIEWER	Joan Webster Royal Brisbane and Women's Hospital Australia
REVIEW RETURNED	12-Dec-2017

GENERAL COMMENTS	Thank you for the opportunity to review your protocol, which provides details of a feasibility RCT comparing a standard dressing with a negative pressure wound dressing to prevent SSI in hip fracture patients requiring surgery. It is an important study and larger than any published study investigating NPWT to date. In addition, it is independently funded (apart from, it seems, some help from S&N for product costs). I have just one suggestion: On page 14 - line 53 You state that the treating surgeons will not be blinded to the treatment. For the primary outcome (page 15, line 53) you state that the treating clinical team will make the diagnosis of 'deep infection'. Perhaps you could state under the heading 'blinding' that outcome assessors will not be blinded to the intervention.
---

REVIEWER	Dr Nicholas D Gollop Norwich Medical School, University of East Anglia, UK.
REVIEW RETURNED	03-Jan-2018

GENERAL COMMENTS	Thank you for inviting me to review your study protocol. I complement the authors on this protocol as it is well written, structured correctly and the study design is appropriate to answer the clinical question. I note that the topic of informed written consent is complex and that there are several issues surrounding this. The authors have correctly addressed this point well and this looks to be appropriate. Exclusion and inclusion criteria all appear reasonable, including the exclusion of percutaneous screw fixation, but I am unclear why only >65year old patients are included and wonder if this is necessary; I suggest taking all-comers would be more valid in a population study. A few small points for attention: 1) Abstract, introduction - reword "precise estimate", it is not possible to achieve both simultaneously, suggest "true estimate". 2) Abstract, Methods - you report follow-up is at 30 days and 4 months. In the main protocol you interchange between '4 months' and '120 days'. Be aware that depending on which months the subjects are recruited, 4 months does not equate to 120 days. I would suggest you offer yourself flexibility where possible. Typographical - please review and remove errors: i) double full stop in limitations ii) remove unnecessary 'the' in paragraph 7 of introduction iii) replace fullstops for comma in randomisation section, second sentence iv) remove additional space at the start of heading of 'rehabilitation' section
---

VERSION 1 – AUTHOR RESPONSE

Reviewer 1 (Michael Whitehouse)

I would prefer to see a standard consent form included as well as the current "Prospective Confirmation of Informed Agreement" form as patients capable of providing informed consent should be given the opportunity to do so and the wording of the form does not really fit for this group.

Response: Agreed and included

In the study design, define that the cohort is prospective.

Response: Changed (line 260).

Cite the latest NHFD report when data from it is cited.

Response: Changed (line 144, 543 and reference number 1). We have kept the reference to the 2015 report as this is where the projection of 100 000 by 2020 comes from.

When discussing the urgency of operative care, please mention the current NICE target of surgery within 36 hours and cite the guidance.

Response: Changed (line 294-295)

It should be acknowledged that the CDC definition of deep infection in the presence of an implant takes place at 1 year following surgery and therefore in this feasibility study, an earlier time point has been chosen and this decision justified.

Response: In 2013 the CDC definition was modified to complete follow up at 90 days for specific (usually implant related) procedures where it was previously 12 months. Repair of neck of femur is included in the list of procedures that should be followed up to 90 days as opposed to 30 days (1).

Adverse event management: The classification of "further surgery to remove/replace metalwork" and "wound dehiscence" as complications that are expected as part of the surgical intervention in the context of a trial with infection as the primary outcome measure is interesting as both of these complications will be strongly correlated with the primary outcome. Whilst I agree with the reporting time line suggested by the authors, careful consideration needs to be given to whether these outcomes are included as primary outcomes, adverse events or complications.

Response: We agree that the complications of "further surgery" and "wound dehiscence" will be closely aligned to the outcome of "infection". Our complications reporting forms which are present in the WHiTE cohort as a whole direct sites completing these forms to review the patients data in light of the outcome of SSI. If this review of data identifies an SSI then it will be included as 'primary outcome'. If it does not then it will be reported complication. We agree that it is important that these specific complications are of high interest to the reader, and we will endeavour to report it as such.

Consideration also needs to be given to the classification of the outcomes of interest given this is a feasibility study design. In the context of a feasibility study, it is usual to consider outcomes such as the ability to recruit to the trial, recruitment rates, and completion of outcome measures etc. as the outcomes of the study. The primary outcome described (deep infection) is actually the primary outcome of the definitive study proposed.

Response: We agree that SSI will be the 'primary outcome' of the main trial should it go ahead. However, as the sample size calculation of which the estimate of the SSI rate is the key unknown input, it is also in this sense the primary outcome of this feasibility study. The prospective use of this outcome in the hip fracture population is novel and its utility in patients with cognitive impairment unknown. We feel this adds a feasibility element to this outcome. Given that whether or not the full scale is 'feasible' will depend on the rate of infection captured in this study. We will report other, perhaps more commonly used feasibility measures, as well.

Statistical analysis - the inclusion of data collected in this feasibility study in the analysis of any subsequent definitive trial suggests that the authors intend to use exactly the same outcome measures. This would be more consistent with this being a pilot study design. This also further emphasises the critical need to justify the decision not to use the standard CDC definition of deep infection in the presence of an implant at 1 year post intervention.

Response: We will only include the feasibility data if the study is deemed feasible and if the feasibility work does not change the design of the study in any way that would challenge the consistency of the results. Our use of 4 months follow up has been clarified in the statistical analysis plan, in line with the updated 90 day final follow up (line 524).

Reviewer: 2 (Joan Webster)

You state that the treating surgeons will not be blinded to the treatment. For the primary outcome (page 15, line 53) you state that the treating clinical team will make the diagnosis of 'deep infection'. Perhaps you could state under the heading 'blinding' that outcome assessors will not be blinded to the intervention.

Response: Changed (line 335)

Reviewer: 3 (Nicholas D Gollop)

Exclusion and inclusion criteria all appear reasonable, including the exclusion of percutaneous screw fixation, but I am unclear why only >65year old patients are included and wonder if this is necessary; I suggest taking all-comers would be more valid in a population study.

Response: The over 65 cut off limit the age of patients included. Younger patients with hip fracture are often high energy injuries, and have fewer co-morbidities. They represent a different cohort of patients with higher energy. This study is concerned with 'fragility type' hip fractures.

A few small points for attention:

1) Abstract, introduction - reword "precise estimate", it is not possible to achieve both simultaneously, suggest "true estimate". (line 52)

Response: We have changed to "value estimated with a greater precision"

2) Abstract, Methods - you report follow-up is at 30 days and 4 months. In the main protocol you interchange between '4 months' and '120 days'. Be aware that depending on which months the subjects are recruited, 4 months does not equate to 120 days. I would suggest you offer yourself flexibility where possible.

Response: Changed so that 4 months is used consistently (line 63, 239, 249, 390, 392, 398, 428, 435, 458, 475, 524)

We hope these responses and revisions address the points raised by the reviewers, and look forward to hearing from you.

Yours sincerely

Dr James Masters on behalf of co-authors

1. Centre for Disease Control.
<https://http://www.cdc.gov/nhsn/pdfs/pscmanual/9pscscsscurrent.pdf> 2018 [cited 2018 19/01/2018].

VERSION 2 – REVIEW

REVIEWER	Michael Whitehouse University of Bristol, UK
REVIEW RETURNED	28-Jan-2018
GENERAL COMMENTS	Thank you to the authors for thoroughly addressing my comments, I am happy with the revised version.
REVIEWER	Nicholas D Gollop Norwich Medical School, UK
REVIEW RETURNED	12-Feb-2018
GENERAL COMMENTS	Thank you for addressing my comments and the comments of my reviewing colleagues. I am satisfied that the protocol is suitable for publication in BMJ Open. Good luck with your study; I would be very interested to hear of the study results.